# Analytical Validation of Two Assays for Equine Ceruloplasmin Ferroxidase Activity Assessment

**DOI:** 10.3390/vetsci10100623

**Published:** 2023-10-18

**Authors:** Stefano Cecchini Gualandi, Tommaso Di Palma, Raffaele Boni

**Affiliations:** Department of Sciences, University of Basilicata, Campus Macchia Romana, 85100 Potenza, Italy; taotaotdp@gmail.com (T.D.P.); raffaele.boni@unibas.it (R.B.)

**Keywords:** ammonium iron(II) sulfate, blood plasma, ceruloplasmin, enzyme, horse, *o*-dianisidine, oxidase activity

## Abstract

**Simple Summary:**

Ceruloplasmin (Cp) is a glycoprotein playing many physiological roles; among them, antioxidant defense is one of the most analyzed, making it an interesting biomarker in animal welfare assessment. Usually, the assessment of this protein is based on an enzymatic assay in which substrate oxidation generates a colored solution whose intensity is related to the Cp content. However, many enzymatic assays commonly developed in humans are applied in veterinary sciences without assessing their species-specific analytical optimal responsiveness. In this paper, two Cp oxidase activity assays using different substrates were tested in the blood plasma of horses and compared on the basis of their analytical reliability. The optimization of these methods for the use in equines was carried out by varying some analytical parameters; among them, the buffer pH is the preeminent variable affecting the analytical output. Our results show that both methods are reliable for the Cp oxidase activity evaluation; nevertheless, the discrepancy observed in the analytical values expressed as Units L^−1^ makes their comparison unfair.

**Abstract:**

Ceruloplasmin (Cp) assessment in biological samples exploits the oxidase activity of this enzyme against several substrates, such as *p*-phenylenediamine (*p*-P), *o*-dianisidine (*o*-D) and, most recently, ammonium iron(II) sulfate (AIS). Once developed in humans, these assays are often used in veterinary medicine without appropriately optimizing in the animal species of interest. In this study, two assays using AIS and *o*-D as substrates have been compared and validated for Cp oxidase activity assessment in horse’s plasma. The optimization of the assays was performed mainly by varying the buffer pH as well as the buffer and the substrate molar concentration. Under the best analytical conditions obtained, the horse blood serum samples were treated with sodium azide, a potent Cp inhibitor. In the *o*-D assay, 500 µM sodium azide treatment completely inhibits the enzymatic activity of Cp, whereas, using the AIS assay, a residual analytical signal was still present even at the highest (2000 µM) sodium azide concentration. Even though the analytical values obtained from these methods are well correlated, the enzymatic activity values significantly differ when expressed in Units L^−1^. A disagreement between these assays has also been detected with the Bland–Altman plot, showing a progressive discrepancy between methods with increasing analytical values.

## 1. Introduction

Ceruloplasmin (Cp) is α-2 glycoprotein synthesized by the hepatic cells [1]. Apart from its well-known role in the transport and metabolism of iron and copper (Cu) in blood, Cp is currently considered a multifunctional protein. Among its many functions, enormous interest has been directed towards its antioxidant activity deriving from the oxidation of iron from the ferrous (Fe^2+^) to the ferric (Fe^3+^) state, with the reduction of molecular oxygen to water. This oxidation allows ferric ions to bind to transferrin; therefore, this phenomenon reduces the availability of these ions to catalyze the degradation of hydrogen peroxide into hydroxyl radicals, thus assuming a detoxifying role [2]. Moreover, Cp, as the foremost protein involved in Cu transport, inhibits the copper ion-stimulated formation of reactive oxidants, responsible for the unbalance between oxidant/antioxidant substances, and myeloperoxidase, an enzyme produced by neutrophils liable for oxidative modifications of organic macromolecules. All this makes Cp essential in antioxidant defense [3,4,5].

Furthermore, CP is part of the Acute Phase Proteins (APPs), highly conserved plasma proteins secreted by the liver or various extrahepatic tissues. Serum concentrations of APPs may significantly increase or decrease in response to several injuries, independently of their location and cause, to favor the systemic regulation of defense, coagulation, proteolysis, and tissue repair (for review see [6]). Hence, Cp evaluation may be used as a diagnostic tool of pathologies or welfare in humans and domestic animals. In humans, alterations of plasma Cp levels can be found in several metabolic and degenerative diseases. Cp levels decrease in degenerative diseases, such as protein synthesis disorders, nephrotic syndrome, Wilson’s disease, and diabetes mellitus [7,8,9,10], whereas Cp increases in many neoplastic and inflammatory disorders, including cirrhosis [11,12,13]. In animals, the assessment of Cp levels has been used as an indicator of inflammation, such as endometritis [14] and mastitis [15] in cows, several inflammatory diseases in pigs [16], in calf diarrhea [17], and in bitch pyometra [18] as well as a metabolic parameter that helps to evaluate copper levels in relation to the lactation stage in dairy cows [19]. Furthermore, a significative relationship has been found between Cp oxidase activity and natural antibody (NAb) level in goat kid serum [20] as well as in dairy cows managed without the dry period [21,22], suggesting a role of this protein in the innate immune system.

In the horse, variations of plasma Cp levels, together with other APPs, have been associated with specific metabolic and infectious diseases, such as obesity [23] and induced acute laminitis [24], as well as endurance racing-induced stress [25]. In clinical practice, blood Cp values are suitable for discriminating therapeutic treatments of colic syndrome during the acute-phase response; in fact, high blood Cp values suggest a surgical resolution, while physiological Cp values suggest a clinical treatment [26]. Interestingly, in inseminated mares, five days post-ovulation, Cp content in uterine fluid increased [27]. This occurrence could be traced to the endometrial preparation for recognition, attachment, and development of the embryo. On the other hand, an increase in Cp levels may also reveal inflammatory events triggered by seminal residues.

Among methods for Cp assessment, the immunoturbidimetric and the enzymatic assays are the most commonly used; however, the latter shows greater analytical sensitivity than the former [28]. These enzymatic assays are based on the oxidation capacity of Cp of non-physiological substrates, such as *o*-dianisidine [29] and *p*-phenylenediamine [30]. The oxidation of these substrates produces a color of the analytical mixture, whose intensity is proportional to the Cp content and readable with a spectrophotometer. Recently, a new automatic method has been proposed by Neşelioğlu et al. [31]. This method uses Cp’s ability to oxidize the ferrous ions to form directly yellow-colored ferric ions; thus, the reaction can be read without a chromogenic reagent.

In veterinary medicine, enzyme activity assays developed in humans are often used without a preliminary adaptation of the method to the species of interest. This can lead to inaccurate results since suboptimal analytical conditions may alter the expression of enzyme activities.

The objective of the present study was to validate and compare two methods that use two substrates, ammonium iron(II) sulfate and *o*-dianisidine, for Cp ferroxidase activity assessment in horse blood plasma.

## 2. Materials and Methods

### 2.1. Animals and Blood Sampling

Ten healthy adult horses of several breeds (i.e., Haflinger, Murgese, and crossbreeds), aged 2 to 15 years and weighing approximately 300 to 450 kg, were enrolled in this study. The horses were housed at a private farm in the Potenza district (Italy) and kept in box stalls with an open paddock in natural light conditions and fed mixed meadow hay ad libitum and 1 kg day^−1^ integrated compound feed for horses (Equimix, Specialmix Miglionico s.r.l., Altamura, Italy). The health status of the horses was assessed on the basis of the anamnestic data as well as on the clinical and haematological analyses to which the horses were routinely subjected. All animals used were client-owned, and informed consent was obtained from the owners.

The horses were blood-sampled only once in resting conditions, in the morning, before feeding. Blood samples were collected from the jugular vein into heparinized vacutainer tubes. Plasma was obtained by centrifugation (2000× *g*, 10 min at 4 °C) and was aliquoted into plastic tubes. Samples were stored at −80 °C until analyses, which occurred within two weeks of sample collection.

### 2.2. Biochemical Assays

Cp oxidase activity was assessed following two methods based on the use of two different substrates: ammonium iron(II) sulfate • 6H_2_O (AIS) and *o*-dianisidine • 2HCl (*o*-D). The first method was performed using a colorimetric assay, as described by Neşelioğlu et al. [31], in which ferrous (Fe^2+^) ions are oxidized to ferric ions (Fe^3+^) to form yellow products whose absorbance can be spectrophotometrically read at 415 nm. In the second method, based on the original assay by Schosinsky et al. [29] and adapted to microplate by Stepien and Guy [32], the chromogen *o*-D is oxidized, in the presence of oxygen, by Cp to form a yellowish-brown mixture that can be measured spectrophotometrically at 550 nm. The buffer acetate pH, the substrate, and the buffer acetate molar concentrations were optimized for both methods. Finally, once the optimal analytical conditions were established, the analyses were carried out in the presence of increasing concentrations of sodium azide, an effective CP inhibitor, up to the inhibition of the analytical signal. Once the plasma samples had been analyzed individually for the optimization of the methods and the analytical signal inhibition by sodium azide, the intra- and inter-assay coefficients of variation (CVs) for each Cp oxidase activity assay were calculated by analyzing two plasma pools with high and low Cp oxidase activity. These pools were assessed five times in a single run and five times on separate days, respectively.

#### 2.2.1. Cp Oxidase Activity Assay Using Ammonium Iron(II) Sulfate (AIS) as Substrate

Plasma samples (45 µL) in duplicate were incubated in a microtiter plate with 150 µL of sodium acetate buffer, prepared by adding acid acetic solution to sodium acetate trihydrate solution, and the mixture was read at 415 nm with reference wavelength of 630 nm using a microplate reader (model 550, BioRad, Segrate, Milan, Italy) against a blank in which saline solution replaced the sample. After the first reading, 20 µL of the substrate was added to each well, and the absorbance of resulting yellow products, as a consequence of the oxidase enzyme activity, was read again after 10 min. The difference between the second and the first reading (ΔAbs) was used to calculate the Cp oxidase activity (Units, U L^−1^), as indicated by the original method [31].

For the analytical optimization, the sodium acetate buffer pH varied from 5.0 to 7.0, the substrate concentration varied from 10 to 50 mM, and the sodium acetate buffer concentration varied from 350 to 550 mM.

Once the best analytical condition was obtained, this was tested with blood plasma samples pre-treated with increasing concentrations of sodium azide (from 0 to 2000 µM).

#### 2.2.2. Cp Oxidase Activity Assay Using *o*-Dianisidine (*o*-D) as Substrate

Plasma samples (20 µL) in duplicate were incubated at 30 °C for 5 min in two separate microtiter plates with 80 µL of sodium acetate buffer (see above). Subsequently, 25 µL *o*-D solution was added into each well on both plates. After 5 and 15 min at 30 °C, 230 µL of 9M H_2_ SO_4_ was added into the wells to stop the enzymatic reaction. The absorbance of the purplish-red solution, resulting from the oxidation of *o*-D, was measured at 550 nm with reference wavelength of 630 nm using the microplate reader. The difference between the second and the first reading (ΔAbs) was used to calculate the Cp oxidase activity (U L^−1^) in terms of consumed substrate, as indicated by Stepien and Guy [32].

For the analytical optimization, the sodium acetate buffer pH varied from 4.2 to 6.7, the substrate concentration varied from 3.15 to 12.61 mM, and the sodium acetate buffer concentration varied from 50 to 200 mM.

Using the optimized analytical conditions, CP oxidase activity was measured in the presence of increasing sodium azide concentrations, as described above.

### 2.3. Statistical Analysis

The analytical data are presented as means ± standard deviation (SD) and consist of the averages of three analyses performed by the same operator. The Kolmogorov–Smirnov and Leven’s tests were used to determine the normality of the distribution and the variance homogeneity of the data.

For the analytical optimization, data obtained by applying the different conditions for the buffer pH and for the molar concentration of substrate and buffer were analyzed by one-way ANOVA. Dunnett’s test was used to compare the best analytical conditions for either buffer pH or substrate and buffer concentrations with the other analytical conditions. ANOVA and Dunnett’s test were also applied to discriminate differences in the analytical recovery values of Cp oxidase activity in samples previously treated with sodium azide in comparison with non-treated (control) samples. Correlation coefficients (r^2^) were calculated by a regression procedure. The minimum level of statistical significance was accepted for *p* < 0.05. All these statistical analyses were performed using SigmaPlot for Windows Version 11.0 statistical software.

A Bland–Altman plot was applied to evaluate the agreement between the two assays and to reveal any possible bias [33] using the open-source software Jamovi (The Jamovi project, Version 2.3.21.0).

The IC_50_ values, i.e., the concentration of sodium azide causing the 50% of inhibition of the analytical signal, were obtained with Microsoft Excel software (Office LTSC Professional Plus 2021).

## 3. Results

### 3.1. Analytical Optimization of the Two Cp Oxidase Activity Assays

Both assays were highly affected by the sodium acetate buffer pH (Figure 1). As regards the AIS assay, the analytical signal increased together with increasing buffer pH up to pH 6.2; beyond this value, the analytic signal decreased. A similar trend was also observed for the *o*-D assay, in which the best analytical signal was reached at pH 5.7.

As regards the molar substrate concentration, the signal peak was found at 20 mM and 7.88 mM in the case of the AIS and *o*-D assays, respectively (Figure 2). In the AIS assay, the analytical signal decreased or increased together with reducing or increasing the substrate concentration. In the *o*-D assay, an increase in substrate concentration beyond the analytical peak value did not significantly affect the Abs values.

The molar concentration of the sodium acetate buffer also significantly affected the analytical values, with the maximum analytical signal observed at 500 mM and 100 mM for the AIS and *o*-D assays, respectively (Figure 3).

### 3.2. Effect of the CP Inhibitor Sodium Azide on the Analytical Signal of the Two Assays

In both assays, the inhibition of the analytical signal of the Cp enzymatic activity resulting from treatment with increasing sodium azide concentrations was evaluated using the optimized analytical conditions, as described above. In particular, in the AIS assay, the acetate buffer pH was set to 6.2, the acetate buffer concentration to 500 mM, and the substrate concentration to 20 mM. In the *o*-D assay, the acetate buffer pH was set to 5.7, the acetate buffer concentration to 100 mM, and the substrate concentration to 7.88 mM. When sodium azide was added to the samples before the execution of the two Cp ferroxidase activity assays, the analytical signals showed a significant progressive decrease together with increasing sodium azide concentration (Figure 4). In addition, 125 µM and 31.25 µM sodium azide were the lowest concentrations at which a significant decrease in analytical signal was observed in the AIS and *o*-D assays, respectively.

The IC_50_ values, expressed as the sodium azide concentration causing the 50% inhibition of the analytical signal, were 386.41 and 41.63 µM for the AIS and *o*-D assays, respectively.

### 3.3. Relation between the Two Cp Oxidase Assays and Precision Analysis

The relationship between the two assays was tested by regression analysis, and a significant (*p* < 0001) correlation coefficient (r^2^) of 0.867 was found (Figure 5A). The Bland–Altman plot showed a progressive discrepancy between these two assays with increasing analytical values (bias = 145.20) (Figure 5B).

Table 1 shows the results of the precision analysis of the two assays. For both methods, plasma samples were divided into two aliquots representing samples with higher or lower Cp oxidase activity. Both the assays returned similar CVs; the intra-assay CVs were lower than the inter-assay CVs for both methods, and all the CVs were lower than 10%.

## 4. Discussion

The acute phase response is a sudden systemic reaction of the organism, commonly an inflammation, to local or systemic disturbances in its homeostasis caused by infection, tissue injury, trauma or surgery, neoplastic growth or immunological disorders [34]. An example of this response was evaluated in the horse by Smith and Cipriano [35], who found a significant increase in Cp oxidase activity following intramuscular administration of turpentine. The evaluation of Cp in biological fluids is a diagnostic tool of growing interest and can be performed using different procedures. Some of them, as assessments of Cp enzymatic activity, do not allow obtaining a precise estimate of the Cp content of a biological sample. In fact, this evaluation is strongly affected by methodological variables, which, in turn, may bias the analytical estimation of protein content. Conversely, other procedures, such as the immunoturbidimetric assay that is based on the reaction between Cp and an anti-Cp antibody, allow the assessment of Cp content that does not always correspond to an analogous oxidase activity of the sample [36]. Therefore, Cp ferroxidase activity assays should be the analyses of choice for the assessment of Cp physiological activity [31].

Various methods have been developed based on the ferroxidase activity of Cp. Most of these assays use polyamine substrates, such as p-phenylenediamine (*p*-P) and *o*-D, and are widely applied. Unlike the previous tests, the AIS test exploits the ability of the protein to oxidize the ferrous ions, which represent a substrate already contained within the sample [37].

An often-underrated aspect of assays measuring enzyme activities is that such tests are applied in several animal species, maintaining the reference assay’s analytical conditions, as developed in humans. This can lead to analytical errors by reducing the analytical power of the test. The pH of the analytical buffer is one of the main variables affecting the activity of enzymes. Indeed, pH affects the efficiency of an enzymatic reaction by varying the charge of functional residues in the binding of the substrate or in the catalysis process and can determine changes in enzymatic conformation [38]. In the Cp enzymatic activity assessment, one of the analytical conditions that mostly influenced the analytical values was the pH of the buffer. All the Cp oxidase activity assays have been developed in humans. So, the optimum pH has been referred to as 5.0, 5.45, and 5.8 for the *o*-D, *p*-P, and AIS assays, respectively [29,30,31]. In animal species in which the analytical conditions have been tested, there is often a mismatch in the values obtained with respect to those indicated in humans. In pigs, the optimum pH is 4.6 and 6.0 by using *o*-D and *p*-P as substrates, respectively, with a progressive decrease up to about zero at pH 4.2 in the *o*-D assay [16]. In dogs, using the *p*-P assay, the optimum buffer pH is 5.2 with a reduction of about 40% of the analytical signal at 5.6 in pyometra-affected animals [18]. In horses, 6.6 is the best pH value using the *p*-P assay, with approximately 40% loss of the analytical signal at pH 5.45, which represents the best pH value for humans [39]. In the present study, our results support the species-specificity of the pH values of the buffer, differing from the pH values of the original methods. Thus, at least in horses, the optimal pH values were 6.2 and 5.7 for the AIS and *o*-D assays, respectively, against 5.8 and 5.0 of the original methods developed in humans. In particular, the *o*-D assay showed a greater reduction in the analytical signal at the pH value set for humans. Regarding the substrate molar concentration, our results agree with those of the original methods described for humans, for both AIS and *o*-D assays. Conversely, in pigs, the optimal substrate concentrations differ from those indicated in the original methods and appear to be significantly lower in either *p*-P or *o*-D assays [16]. Furthermore, in dogs, the optimal substrate concentration in the *p*-P assay is approximately half of the original human method, with the optimal substrate concentration higher in pathological than in healthy specimens [18]. Concerning the molar concentration of the acetate buffer, our results agree with that of the original method for the *o*-D assay; however, a higher buffer concentration (500 mM) seems to ensure a better analytical signal in the AIS assay with respect to that used in humans, set at 450 mM.

The present results showed that when plasma samples were treated with sodium azide, a potent inhibitor of Cp, the analytical signal was progressively reduced in either *o*-D or AIS assays. Sodium azide is an inorganic compound able to bind copper, inhibiting Cp oxidase activities [37], as previously demonstrated [40,41,42]. Using sodium azide, we were able to discriminate these two methods based on the different response obtained. In fact, using the *o*-D assay, a complete inhibition was achieved at 500 µM sodium azide, with the IC_50_ at 41.63 µM. In contrast, using the AIS assay, a complete inhibition did not occur even at the highest sodium azide concentration (2000 µM), with a residual analytical recovery of about 18% and an IC_50_ at 386.41 µM. Thus, in the latter case, a possible interference of some plasma molecules, reacting with ferrous ions of the substrate and determining a residual analytical signal may be hypothesized. Actually, even in humans, using similar treatments, the analytic signal is not thoroughly annulled. In the AIS original method, the authors found that sodium azide inhibits Cp oxidase activity at the level of 95–99%; unfortunately, they did not report the tested concentration [31] and, hence, a comparison with our results is unreliable.

Reporting analytical data as U L^−1^ and comparing these with literature data, horses show significantly lower Cp oxidase activity than humans. Horse CP enzyme activity is 167.02 ± 33.43 and 21.83 ± 5.10 U L^−1^ by AIS and *o*-D assays, respectively, while in healthy humans, the Cp oxidase activity is 724 U L^−1^ by the AIS assay [31] and 76 U L^−1^ by the microplate *o*-D method [32]. Surprisingly, it is impossible to compare data obtained with these methods. Indeed, in either our study or the reference literature, the Cp oxidase activity is about eight to ten times higher in the AIS assay than in *o*-D assay-analyzed samples. Even if the analytical data are significantly correlated, their absolute values largely differ, as also revealed by the Bland–Altman plot, which highlights a progressive discrepancy between the two assays (bias = 145.20, Figure 5B) at increasing Cp levels. Similar discrepancies also occur using other assays. In healthy human samples, for example, the Cp oxidase activity is evaluated as approximately 100 U L^−1^ by using the *o*-D method [7] but more than double by using the *p*-P assay [43]. In animals, discrepancies in the Cp enzyme assessment were previously observed by Martínez-Subiela et al. [16] in the validation of the Cp oxidase activity in pigs using either *p*-P or *o*-D as substrate, as well as by Hussein et al. [44], comparing the manual and automated methods for the bovine Cp enzyme activity using *p*-P as substrate. Thus, the comparison of analytical data from different methods is incorrect.

## 5. Conclusions

Two methods for assessing Cp oxidase activity have been validated for horse blood plasma. The main difference between these two tests is related to the substrate used. For both methods, the main methodological variable affecting analytical values is the pH of the buffer. This must be taken into consideration when enzymatic assays are used in species other than those in which the test has been developed. Even though the results of these assays are related, the discrepancy observed in the analytical values of the two methods when expressed as U L^−1^ highlights the impossibility of comparing the analytical data obtained between different assays. Once this methodological optimization phase conducted on blood samples of healthy horses has been overcome, future experiments on horses affected by specific pathologies will discriminate the reliability of these two assays in detecting alterations in Cp activity.

## Figures and Tables

**Figure 1 vetsci-10-00623-f001:**
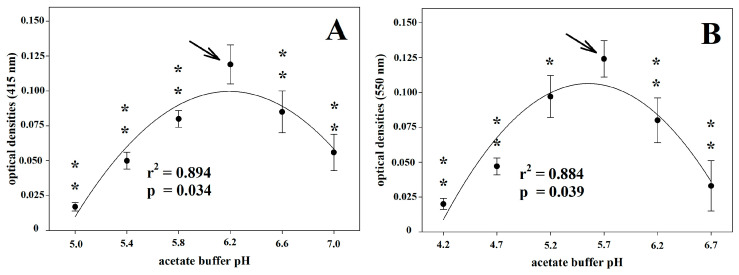
Optimum pH of the acetate buffer for ammonium iron(II) sulphate assay (**A**) and for *o*-dianisidine assay (**B**). One or two asterisks indicate statistically significant (*p* < 0.05 or *p* < 0.01, respectively) differences in comparison to the best analytical condition that is indicated by the arrow.

**Figure 2 vetsci-10-00623-f002:**
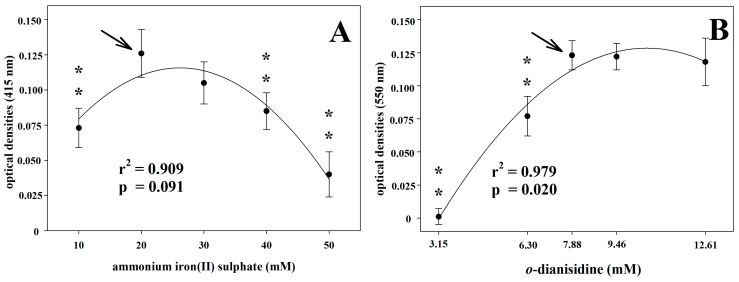
Optimum substrate molar concentration for ammonium iron(II) sulphate assay (**A**) and for *o*-dianisidine (**B**). Two asterisks indicate statistically significant (*p* < 0.01) differences in comparison to the best analytical condition that is indicated by the arrow.

**Figure 3 vetsci-10-00623-f003:**
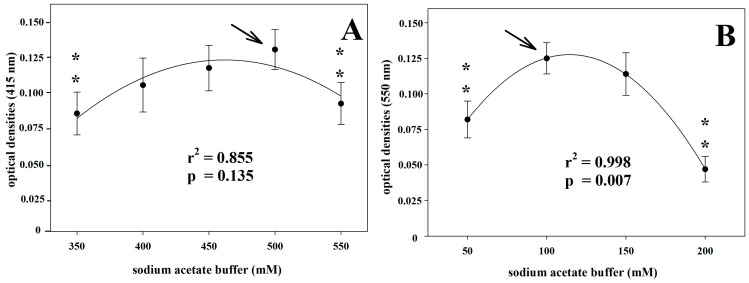
Optimum molar concentration of the acetate buffer for ammonium iron(II) assay (**A**) and for *o*-dianisidine assay (**B**). Two asterisks indicate statistically significant (*p* < 0.01) differences in comparison to the best analytical condition that is indicated by the arrow.

**Figure 4 vetsci-10-00623-f004:**
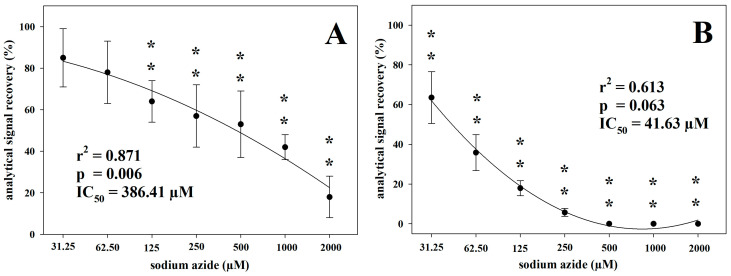
Analytical signal recovery in sodium azide-treated samples in comparison to the untreated control for the ammonium iron(II) sulphate (**A**) and *o*-dianisidine (**B**) assays. Two asterisks indicate statistically significant (*p* < 0.01) differences in comparison to the untreated control.

**Figure 5 vetsci-10-00623-f005:**
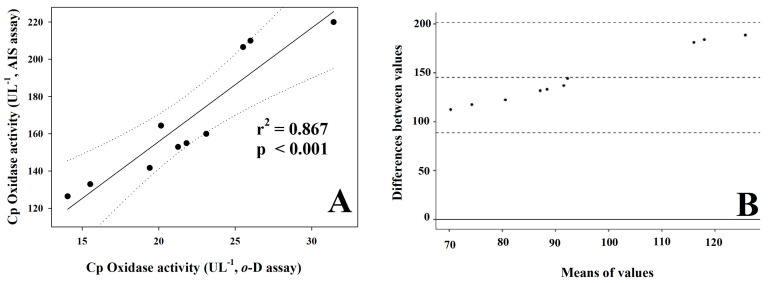
Regression analysis (**A**) and Bland-Altman plot (**B**) between analytical values of the two methods, presented as U L^−1^. Plot A: values obtained with the two methods were plotted (full circles) and adjusted to a regression line (black line); the dashed lines represent the 99% confidence interval curves. Plot B: *y*-axis: differences between the values obtained with AIS and *o*-D assays; *x*-axis: mean of the Cp oxidase activity values obtained with the two methods. The solid line indicates the zero-bias line, i.e., where the differences equal to zero are placed; the central dashed line represents the mean differences between the measurements of the two methods (bias); the two dotted lines at the top and bottom represent the 95% limits of agreement (mean ± 1.96 SD).

**Table 1 vetsci-10-00623-t001:** Precision analysis of ceruloplasmin (Cp) of AIS and *o*-D assays.

	Intra-Assay		Inter-Assay	
	Mean ± SD	CV (%)	Mean ± SD	CV (%)
AIS assay (U L^−1^)				
High Cp samples	200.24 ± 3.72	1.86	210.78 ± 15.43	7.32
Low Cp samples	144.87 ± 2.88	1.99	160.37 ± 12.24	7.63
*o*-D assay (U L^−1^)				
High Cp samples	26.52 ± 0.65	2.46	25.83 ± 1.51	5.86
Low Cp samples	18.71 ± 0.43	2.31	18.11 ± 0.88	4.89

## Data Availability

The data presented in this study are available on request from the corresponding author.

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
