# Peer review of "Analytical Validation of Two Assays for Equine Ceruloplasmin Ferroxidase Activity Assessment"

_vetsci, 2023, doi:10.3390/vetsci10100623_

Round 1

Reviewer 1 Report

This study is based on a good idea but the execution is sub-par. The sample size of the study is too low to determine the agreement between the 2 methods.

What is the significance of Cp in horses specifically? Why did the author choose horses only? A similar study has been done on cows and other animals. For the validation of 2 methods a multi-species system would be a much better approach. Only using one species and with a low sample size is a big negative for this study. 

Bland-Altmann plots are not enough for agreement analysis. A correlation curve is also required.

The quality of English is average. 

Author Response

This study is based on a good idea but the execution is sub-par. The sample size of the study is too low to determine the agreement between the 2 methods.

What is the significance of Cp in horses specifically?

R. We thank the Reviewer very much for this suggestion which allowed us to improve the readability of our paper. The required information has been added in the text.

Why did the author choose horses only? A similar study has been done on cows and other animals. For the validation of 2 methods a multi-species system would be a much better approach. Only using one species and with a low sample size is a big negative for this study.

R. The choice to consider only the equine is aimed at filling the lack of information on the use of these two methods in this species. The optimization of these methods resulted in a considerable amount of analysis that is shown in ten graphs. A multi-species comparative study is undoubtedly worthy of attention and may indeed represent a hint for further experiments. However, optimizing only one analytical method at a time would be more appropriate. As regards the sample size, we would like to underline that our study was aimed at optimizing analytical methods in a species rather than providing species-specific reference values. This can be achieved with a low number of subjects, as shown by other authors in similar papers (doi:10.1016/j.rvsc.2006.10.011; 10.1051/vetres:2004046). We also believe that the analyses done on 10 individuals do not correspond to a low number of cases. Conversely, it is important to increase the case number for evaluating an analytical range for a given species and discriminating pathological and physiological conditions. This will be the next step of our investigations.

Bland-Altmann plots are not enough for agreement analysis. A correlation curve is also required.

R. Figure 5 shows both the regression line with Pearson’s correlation coefficient (A) and the Bland-Altman plots (B).

We are sure that by resolving the critical issues raised by the reviewer we have improved the quality of our manuscript.

Reviewer 2 Report

I must admit that I am not an expert in this kind of study, and therefore do not feel very confident to critique that scientific approaches used, however I do feel that I can comment on the overall presentation of the manuscript.

This manuscript is well presented and organized, and is easy to understand. The statistical methods used seem appropriate in addressing the questions, and the hypothesis is clear.

In the Methods section 2.2: am I understanding that plasma from the 10 horses was pooled for all experiments, and that 5 replicates were used for each assay? If yes, consider rewording a bit to emphasize that this approach was used for all assays.

I do have a comment regarding Figure 4: consider including the data from the control in the figure, so that the reader can better understand the comparisons made.

Author Response

I must admit that I am not an expert in this kind of study, and therefore do not feel very confident to critique that scientific approaches used, however I do feel that I can comment on the overall presentation of the manuscript.

This manuscript is well presented and organized, and is easy to understand. The statistical methods used seem appropriate in addressing the questions, and the hypothesis is clear.

R. Thanks a lot to the reviewer for her/his positive comments and suggestions that helped us to improve our manuscript.

In the Methods section 2.2: am I understanding that plasma from the 10 horses was pooled for all experiments, and that 5 replicates were used for each assay? If yes, consider rewording a bit to emphasize that this approach was used for all assays.

R. We apologize for any lack of clarity; however, during the optimization of the methods and the analytical recovery after sodium azide treatment, the plasma samples were analyzed as individual specimens. This is shown in Figure 5 (Linear regression line and Bland-Altman plot), where ten points represent the ten samples analyzed. In the case of the intra- and inter-assay coefficients of variation, the samples were analyzed in two pools depending on their Cp activity (high and low). We have added a new sentence in the manuscript to better clarify this part. 

I do have a comment regarding Figure 4: consider including the data from the control in the figure, so that the reader can better understand the comparisons made.

R. Unfortunately, since a logarithmic scale has been used, it is not feasible to add the analytical recovery value at 0 µM sodium azide in the graph because it will be considered 100%.

Reviewer 3 Report

Dear authors, the topic of your work “Analytical validation of two assays for equine ceruloplasmin ferroxidase activity assessment” it is very scientific importance in equine physiology as welfare evaluation like a bioindicator. 

I would like the following considerations to the authors:

Simple summary that is very clear. 

Abstract is ok. 

The introduction is too brief, and I would focus more on the context of the article's subject matter and its applications. In paragraph “Based on its multiple functions…” lines 52 to 64, I suggest more information about the clinical report or applications on equine medicine. 

Material and Methods are very well structured and they are clear. 

The animals measured are understood to be to establish some kind of physiological range. And that they do not have any pathology that may alter the results. This should be better specified. It is suggested in further studies, to make a control group without pathologies compared with some pathology in which the measurement is useful.

 There is a well statistical analysis. 

The results are clearly presented and are accompanied by graphs and supplementary materials.

The discussion of the results is good. 

Conclusions should focus not only on the comparison between the analytical methods, but, if there is one of the two as preferable on the basis of the results with some kind of standard.

In my opinion it is a good paper but it lacks emphasis on its clinical applicability and not only on analytical methodology, but it would fit better in a journal focused on clinical biochemistry.

Author Response

Dear authors, the topic of your work “Analytical validation of two assays for equine ceruloplasmin ferroxidase activity assessment” it is very scientific importance in equine physiology as welfare evaluation like a bioindicator. 

R. Thanks a lot to the reviewer for her/his positive comments and suggestions that helped us to improve our manuscript.

I would like the following considerations to the authors:

Simple summary that is very clear. 

Abstract is ok. 

The introduction is too brief, and I would focus more on the context of the article's subject matter and its applications. In paragraph “Based on its multiple functions…” lines 52 to 64, I suggest more information about the clinical report or applications on equine medicine.

R. The required information has been added to the text. Thanks for the suggestions.

Material and Methods are very well structured and they are clear. 

The animals measured are understood to be to establish some kind of physiological range. And that they do not have any pathology that may alter the results. This should be better specified. It is suggested in further studies, to make a control group without pathologies compared with some pathology in which the measurement is useful.

R. Thanks for the suggestion. A sentence has been added in the M&Ms to better specify the enrollment of healthy animals. In future studies, we are going to verify the reliability of these two methods under pathological conditions.

There is a well statistical analysis. 

The results are clearly presented and are accompanied by graphs and supplementary materials.

The discussion of the results is good. 

Conclusions should focus not only on the comparison between the analytical methods, but, if there is one of the two as preferable on the basis of the results with some kind of standard.

R. As above, we believe that future experiments also involving diseased animals will clarify the choice for using these methods. A sentence in this regard has been added in the Conclusions.

 In my opinion it is a good paper but it lacks emphasis on its clinical applicability and not only on analytical methodology, but it would fit better in a journal focused on clinical biochemistry.

R. We are sure that by resolving the critical issues raised by the reviewer we have improved the quality of our manuscript that may be worthy to be published in Veterinary Science.

Reviewer 4 Report

Good paper be published as submitted

Author Response

Good paper be published as submitted

R. Many thanks to the reviewer for his positive comment on our manuscript

Reviewer 5 Report

Dear Authors,

Thanks for your work "Analytical validation of two assays for equine ceruloplasmin ferroxidase activity assessment". Overall, I think it was well written ,and your results would contribute to consider on which assay would differ in the ceruloplasmin activity analysis in equines.

Please find my suggestions below,

Kind regards,

Reviewer

Summary

Line16: please add “for the use in equines” after > The optimization of these methods for the use in equines was carried out”

Abstract

Line 25: oxidase activity assessment in “horse’s plasma”

Introduction:

From Line 57 and 64 authors describe the interest on analysing this protein in other domestic species. Since they are describing the “Analytical validation of two assays for equine ceruloplasmin ferroxidase activity assessment”. It would be worthy if they add a short paragraph or a few sentence/lines about the use or analysis of this ceruloplasmin in the horse, what does it indicates, what is it use for? What is the implication or interest on clinical analysis of this animal? As well as in the discussion part, a few lines to discuss about what is found in horses’ assays. In the conclusion as well, what would expect to beneficiate the equines from this two types of analysis.

Line 60: please change on…>metabolic parameter “that helps” instead of “parameter helpful

Material and methods

Line 86: ad libitum please in italics format

Results

Line 163: please erase “the” Both the analysis

I think the English writing is understandable, just a few remarks that I think it could improve some parts of the text.

Author Response

Dear Authors,

Thanks for your work "Analytical validation of two assays for equine ceruloplasmin ferroxidase activity assessment". Overall, I think it was well written ,and your results would contribute to consider on which assay would differ in the ceruloplasmin activity analysis in equines.

Please find my suggestions below,

Kind regards,

Reviewer                                 

R. Many thanks to the reviewer for her/his positive comments on our manuscript and for suggestions that helped to improve our manuscript.

Summary

Line16: please add “for the use in equines” after > The optimization of these methods for the use in equines was carried out”

R. done

Abstract

Line 25: oxidase activity assessment in “horse’s plasma”

R. done

Introduction:

From Line 57 and 64 authors describe the interest on analysing this protein in other domestic species. Since they are describing the “Analytical validation of two assays for equine ceruloplasmin ferroxidase activity assessment”. It would be worthy if they add a short paragraph or a few sentence/lines about the use or analysis of this ceruloplasmin in the horse, what does it indicates, what is it use for? What is the implication or interest on clinical analysis of this animal? As well as in the discussion part, a few lines to discuss about what is found in horses’ assays. In the conclusion as well, what would expect to beneficiate the equines from this two types of analysis.

R. We thank the reviewer for this suggestion which was really appreciated. We added information about the Cp significance in horses and its implication in clinical analysis. Furthermore, a final sentence in this regard has been added in the Conclusions.

Line 60: please change on…>metabolic parameter “that helps” instead of “parameter helpful”

R. done

Material and methods

Line 86: ad libitum please in italics format

R. done

Results

Line 163: please erase “the” Both the analysis

R. done

I think the English writing is understandable, just a few remarks that I think it could improve some parts of the text.

R. We revised the entire text to improve the English form.

Round 2

Reviewer 1 Report

The authors have worked on the previous review suggestions. The manuscript reads much better. I recommend the manuscript be accepted.